

# The effect of the dispersal kernel on isolation-by-distance in a continuous population

Tara N. Furstenau and Reed A. Cartwright

School of Life Sciences and the Biodesign Institute, Arizona State University, Tempe, AZ, United States of America

## ABSTRACT

Under models of isolation-by-distance, population structure is determined by the probability of identity-by-descent between pairs of genes according to the geographic distance between them. Well established analytical results indicate that the relationship between geographical and genetic distance depends mostly on the neighborhood size of the population which represents a standardized measure of gene flow. To test this prediction, we model local dispersal of haploid individuals on a two-dimensional landscape using seven dispersal kernels: Rayleigh, exponential, half-normal, triangular, gamma, Lomax and Pareto. When neighborhood size is held constant, the distributions produce similar patterns of isolation-by-distance, confirming predictions. Considering this, we propose that the triangular distribution is the appropriate null distribution for isolation-by-distance studies. Under the triangular distribution, dispersal is uniform over the neighborhood area which suggests that the common description of neighborhood size as a measure of an effective, local panmictic population is valid for popular families of dispersal distributions. We further show how to draw random variables from the triangular distribution efficiently and argue that it should be utilized in other studies in which computational efficiency is important.

## INTRODUCTION

For many populations, individuals do not exist in discrete patches or demes; instead they are spread across a continuous landscape. Although there are no barriers separating individuals, dispersal distances are often limited, and individuals that are near one another in space will be more similar genetically than individuals further apart. This phenomenon is known as isolation-by-distance and introduces a spatial component that should be considered when studying population genetic processes (*Jongejans, Skarpaas & Shea, 2008*). Unfortunately, incorporating multiple dimensions of space at fine scales into analytical models is often analytically intractable (*Epperson et al., 2010*). Therefore, many researchers have turned to spatially-explicit, individual-based computer simulations which offer a more flexible way to incorporate spatial complexity into biological models (e.g., *Barton et al., 2013*; *Cartwright, 2009*; *Epperson, 2003*; *Novembre & Stephens, 2008*; *Rousset, 2004*; *Slatkin, 1993*).

Corresponding author
Reed A. Cartwright,
cartwright@asu.edu

A dispersal kernel describes the distribution of Euclidean distances between birth site and reproduction site. Ideally, when modeling dispersal, the dispersal distribution would be selected based on how well it fits the dispersal kernel estimated from natural populations. Classically, dispersal has been modeled as a diffusion process with Gaussian displacement; however, the observed dispersal kernels in many species tend to be more leptokurtic with a higher probability of short and long distance dispersal (*Bateman, 1950*). In plants, the shape of the dispersal kernel near the origin depends on the mechanism of dispersal; for example, there may be a high peak near the origin for gravity or animal dispersal whereas there may be a minimum near the origin for wind dispersal (*Barluenga et al., 2011*; *Clark et al., 2005*).

The shape of the dispersal kernel impacts many population processes including the rate of population expansion (*Clark, Lewis & Horvath, 2001*; *Kot, Lewis & Van den Driessche, 1996*), responses to environmental changes (*Nathan et al., 2011*), local adaptation (*Berdahl et al., 2015*), speciation (*Hoelzer et al., 2008*), and the spatial distribution of genetic diversity (*Bialozyt, Ziegenhagen & Petit, 2006*; *Ibrahim, Nichols & Hewitt, 1996*). Fat-tailed dispersal kernels, with a higher probability of long-distance dispersal, are a good fit to many empirical data sets (*Bullock & Clarke, 2000*; *Clark et al., 2005*; *Gonzàlez-Martìnez et al., 2006*; *Martìnez & Gonzàlez-Taboada, 2009*; *Klein et al., 2006*). Many studies have shown that population models behave differently when fat-tailed dispersal distributions are used instead of Gaussian dispersal. *Kot, Lewis & Van den Driessche (1996)* demonstrated that population spread is sensitive to the shape of the dispersal kernel and models using a normal distribution underestimated the rate of invasion compared to fat-tailed distributions. *Nathan et al. (2011)* found that long distance dispersal plays a large role during range shifts of wind-dispersed trees in response to projected climate changes. *Houtan et al. (2007)* showed that heavy tailed dispersal kernels were a better fit for dispersal of Amazonian birds but the shape of the dispersal kernel can change in response to forest fragmentation.

While the shape of the dispersal kernel impacts many population processes at different scales, it remains unclear how it effects patterns of isolation-by-distance within a continuous population. It has been argued that the number of long-distance dispersal events will not have a noticeable effect because new long-distance alleles are more likely to be lost due to drift than become established at the new location (*Epperson, 2007*; *Ibrahim, Nichols & Hewitt, 1996*). On the other hand, the shape of the dispersal kernel near the origin may have a significant impact on the overall rate of migration. In plants, this could result in a higher probability of self-fertilization and/or a reduction in the number of successful offspring when there is density dependent regulation (*Barluenga et al., 2011*; *Howe, Schupp & Westley, 1985*; *Moyle, 2006*).

Isolation-by-distance theory predicts that the probability of identity-by-descent between two neutral genes will decrease as the geographic distance between them increases and this pattern can help quantify spatial genetic structure. The analytical model developed by *Malécot (1969)* depends on the effective population density, the mutation rate, the spatial dimensions of the population, and the dispersal distribution. Much of the isolation-by-distance work has focused on the lattice model which forces a constant population

density (*Malécot, 1969*; *Maruyama, 1970*; *Sawyer, 1977*) but these results hold when considering continuously distributed populations with spatial clustering (*Barton et al., 2013*). In two dimensions, the relationship between the probability of identity-by-descent and the log of distance is linear over a certain range of distances and the relationship is proportional to $1/(D_e\sigma^2)$ where $D_e$ is the effective population density and $2\sigma^2$ is the mean squared distance of dispersal (i.e., non-central second moment of Euclidean distance; *Barton et al., 2013*; *Malécot, 1969*; *Rousset, 1997*; *Rousset, 2004*; *Wright, 1946*). Over this range, the slope of the probability of identity-by-descent function is independent of most aspects of the dispersal distribution except for $2\sigma^2$; however, when the distance between individuals falls below the range, the shape of the dispersal distribution becomes important (*Rousset, 1997*). This suggests that as long as $2\sigma^2$ stays constant, any dispersal distribution will produce similar patterns of isolation-by-distance. However, Rousset argues that the magnitude of genetic differentiation will always depend on the shape of the distribution (*Rousset, 1997*; *Rousset, 2008*).

Despite the increase in the use of spatially explicit simulations in studies of spatial genetic structure, it remains unclear whether the shape of the dispersal kernel should be considered. There has not been a clear comparison of how the shape of different dispersal kernels affect observable patterns of isolation-by-distance in these simulations. Here we attempt to offer such a comparison using a spatially-explicit, individual-based model to simulate local dispersal in a continuous population to determine if patterns of isolation-by-distance vary based on the shape of several different dispersal distributions: Rayleigh, half-normal, exponential, triangular, gamma, Lomax, and Pareto. Each dispersal distribution has a different shape, but they can be parameterized such that their non-central second moment is $2\sigma^2$. If the simulations reveal a similar pattern of isolation-by-distance across all dispersal distributions, we can conclude that, for a wide range of dispersal distributions, $2\sigma^2$ is the main determining factor of how genetic similarity declines with increasing distance in a continuous population. Consequently, when designing isolation-by-distance simulations, researchers may choose a dispersal distribution based on computational needs instead of biological fit.

*Wright (1946)* uses the term "neighborhood" to describe a local population from which parents are randomly drawn. He measures the magnitude of the effective size of the neighborhood, $N_b$, as the inverse of the probability that two gametes at the same location came from the same parent. Assuming dispersal is normally distributed along each axis, he calculated that $N_b = 4\pi\sigma^2 D_e$, where $D_e$ is the effective density of individuals, and $2\sigma^2$ is the mean squared distance of dispersal. In his model this captures 86.5% of parents of central individuals. Although Wright assumed Gaussian dispersal, his formula can be used to calculate $N_b$ for many different dispersal models at equilibrium due to the central-limit theorem. $N_b$ is important because it helps define the rate of decay of genetic similarity over spatial distance, i.e., the amount of isolation-by-distance in a population (*Barton et al., 2013*; *Rousset, 1997*; *Rousset, 2000*).

If a neighborhood is supposed to represent a local panmictic unit, then in the ideal model parents should be chosen uniformly from within a circle of radius $2\sigma$ centered on an offspring, and the Euclidean distance between parents and offspring should follow a

triangular distribution: $f(r; \sigma) = r/(2\sigma^2)$, where $2\sigma^2$ is again the non-central second moment. This type of neighborhood is similar to the neighborhood defined in the spatially continuous $\Lambda$-Fleming-Voit disc model in which a number of parents, $v$, are chosen uniformly at random from a disc with radius $r$ to replace a fraction $u$ of the population (*Barton et al., 2013*). In this model, neighborhood size is defined by the ratio $v/u$ and the individuals occupying the disc constitute a panmictic population. If 100% of the population is replaced ($u = 1$), the definition of neighborhood size reduces to the number of individuals competing for the central location.

Below, we demonstrate that patterns of isolation-by-distance are equivalent for different dispersal kernels with the same second moment, and discuss the use of the triangular distribution to model dispersal in a continuous population.

## METHODS

### Simulation

In our individual-based simulation, a population exists on a $100 \times 100$ rectangular lattice (cf. *Epperson, 1995*; *Epperson & Li, 1997*; *Epperson, 2007*; *Hardy & Vekemans, 1999*). Individuals are uniformly spaced with a single individual per cell. Each individual contains one haploid locus. The initial population of 10,000 individuals each carry a unique allele. Generations are discrete, and individuals reproduce by generating 15 clonal offspring that experience mutations according to the infinite alleles model at rate $\mu$. All starting and mutant alleles are selectively neutral.

The offspring disperse from the parent cell following a given dispersal distribution. The landscape boundaries are absorbing, and when offspring disperse off of the lattice they are lost. Offspring that land in the same cell will compete to become a parent in the next generation. Because all alleles are selectively neutral, a single successful offspring is uniformly selected for each cell. To avoid storing all the offspring in memory until dispersal is completed, we use a reservoir sampling method to immediately accept or reject offspring when they land on a cell (*Vitter, 1985*). This method allows us to keep track of two randomly chosen offspring per cell. The first offspring becomes a parent in the next generation and the second individual is recorded to measure the probability of identity-by-descent for offspring competing for the same cell. While it is possible for a cell to remain empty after dispersal, we determined that when each parent generates 15 offspring the number of empty cells per generation is negligible so we assume a constant homogeneous population density.

### Modeling Dispersal

The simulation is spatially-explicit with space represented on a rectangular lattice. Due to the discrete nature of the lattice, the dispersal kernels will be discretized approximations of continuous distributions (*Chesson & Lee, 2005*; *Chipperfield et al., 2011*). The dispersal kernel function, $f(r, \theta; \sigma)$, takes a parameter $\sigma$ and returns continuous polar coordinates. The $\sigma$ parameter is the square root of one-half the second moment of dispersal distance. The polar coordinates include the angle, $\theta \in [0, 2\pi]$, which is uniformly distributed to ensure isotropic dispersal and distance, $r > 0$, which is drawn from a continuous distribution.

Once the angle and distance are drawn, the final position is determined by converting the polar coordinates into rectangular coordinates and adding them to the parent's position. The new coordinates are then rounded to determine the integer coordinates of the destination cell. This dispersal scheme is similar to the centroid-to-area approximation of continuous dispersal kernels described by *Chipperfield et al. (2011)*, which showed minimal deviation from expectations especially when cell length is less than the expected distance.

We looked at seven different dispersal distance kernels (Fig. S1): Rayleigh, exponential, half-normal, triangular, gamma, Pareto, and Lomax. We chose these distributions because they provide a range of shapes for short, intermediate, and long distance dispersal.

The Rayleigh is a distribution of Euclidean distances that result from bivariate normal displacement along the $x$ and $y$ axis. The Rayleigh distribution follows the assumptions of *Wright (1946)*'s two-dimensional isolation-by-distance model.

The exponential distribution is more leptokurtic with higher probability of dispersal at short and long distances and less at intermediate distances. The exponential tail is the boundary that separates truly heavy-tailed distributions with potentially infinite higher moments from distributions with all moments finite. The distinction is important because leptokurtic, heavy-tailed dispersal kernels are typically a better fit to observed dispersal in nature (*Clark, 1998*).

The half-normal distribution is equivalent to a normal distribution that has been folded over the $y$-axis. In this case, Euclidean distance is simply the absolute value of normally distributed random variables. The half-normal is a monotonically decreasing distribution with a convex shoulder near zero. This distribution has a higher probability of dispersal at intermediate distances compared to the exponential.

The triangular distribution is typically defined using three points: a lower limit, $a$, an upper limit, $b$, and a mode, $c$. Here we use a special case of the triangular distribution where $a = 0$ and $b = c = 2\sigma$. We chose this special case of the triangular distribution because in our dispersal function it will return polar coordinates that are uniformly sampled from within a circle with area $4\pi\sigma^2$ which is the same as the neighborhood area (see proof in Appendix A). The triangular distribution is also the only one of our distributions that has a finite range, $r \in [0, 2\sigma]$.

Unlike the previous single parameter distributions, the final three distributions have an additional $\alpha$ shape parameter. The gamma distribution is equivalent to the exponential distribution when $\alpha = 1$, and as $\alpha$ increases the distribution becomes more symmetrical with a higher probability for intermediate distances and a lower probability for short distances.

The Lomax and Pareto distributions are both heavy-tailed power-law distributions. The $n$-th moments are finite only when $\alpha > n$. The support for the Pareto distribution, $r \in [x_{\min}, +\infty)$, begins at a parameter $x_{\min} > 0$. The Lomax distribution is a Pareto distribution that has been shifted so that the support begins at zero. We chose values of $\alpha$ between 2 and 3 so that the second moment of the distribution would be finite but higher moments are infinite.

The dispersal function is executed over 100-billion times per simulation, and thus it is important to make the implementation as efficient as possible. With this aim, we used an xorshift algorithm for uniform pseudo-random number generation and the ziggurat rejection sampling algorithm when applicable (*Marsaglia & Tsang, 2000b*; *Marsaglia, 2003*). We used two different versions of the ziggurat algorithm to draw distances from the exponential and half-normal distribution. For the gamma distribution we used a rejection sampling method that uses the ziggurat algorithm to draw normal variates (*Marsaglia & Tsang, 2000a*). Random variables from the Pareto distribution are generated by $x_{min}e^{X/a}$ where $X$ is an exponentially distributed random variable that we draw using the ziggurat algorithm. The Lomax distribution is sampled similarly: $x_{min}e^{X/a} - x_{min}$.

In addition to generating random distances, the dispersal function requires costly conversions from polar to Cartesian coordinates. We were able to avoid this conversion for the Rayleigh and triangular distributions. We simulated the Rayleigh distribution by drawing vertical and horizontal offsets from independent normal distributions using the ziggurat algorithm. For the triangular distribution, we developed a discrete sampling algorithm using the alias method that allows us to draw the vertical and horizontal offsets simultaneously in constant time (*Vose, 1991*). See Appendix D for a description of the algorithm, and Appendix E for an analysis of its superior efficiency compared to the other distributions we used.

## Analysis

A simulation was run for each of the seven types of dispersal distributions under 4 levels of dispersal ($\sigma = 1, 1.5, 2$ and $4$) with a mutation rate of $\mu = 10^{-4}$. Each simulation was run for a burn-in period of 10,000 generations to allow the population to reach a mutation-drift equilibrium. After the burn-in, data was collected from populations that were 1,000 generations apart to decrease the correlation between them for a total of 2,000 replicate populations per simulation. In each population, a straight transect of 50 individuals was sampled from the center of the landscape to avoid measuring edge effects.

From the transect, all possible pairs of individuals were placed into distance classes based on the geographical distance between the pair. The number of pairs that shared an identical allele was determined and recorded as a proportion of the total number of pairs in the distance class. The probabilities for each distance class were then averaged over all sampled populations. Under this sampling scheme, the number of pairs per distance class decreases as distance increases so in distance class 50 there is only one pair sampled per population.

The parameters for each dispersal distribution were calculated so that $E[X^2] = 2\sigma^2$; the calculations are reflected in the probability distribution functions in Fig. S1. Due to the discrete nature of the lattice, some parameters values were adjusted slightly until the simulations produced an average, observed, squared distance between parent and offspring, $s^2$, that was within 5% of the expected value, $\sigma^2$. Three of the distributions require a second $\alpha$ parameter. For the gamma distribution we used $\alpha = 1, 2, 4, 8$. For the Lomax and Pareto distributions we used $\alpha = 2.4, 2.6, 2.8, 3.0$ all of which are infinite in the 3rd and higher moments.

Under isolation-by-distance, individuals geographically near one another will tend to be genetically similar, and this similarity will decrease as the distance between pairs of individuals increases. Therefore, isolation-by-distance is described by constructing correlograms of genetic similarity between individuals versus the distance between them. Genetic similarity can be measured using identity-by-descent, identity-by-state, relatedness, conditional kinship, or F-coefficients and can be based on coalescent times, an ancestral population, or the current population (*Hardy & Vekemans, 1999*; *Hardy, 2003*; *Malécot, 1969*; *Rousset, 1997*; *Rousset, 2002*; *Wang, 2014*). For two-dimensional populations, genetic similarity is often plotted against the log-distance separating pairs because theory predicts that this relationship is approximately linear (*Barton et al., 2013*; *Hardy & Vekemans, 1999*; *Rousset, 2000*).

We recorded the probability of identity-by-descent for pairs of individuals in each distance class. Under the infinite alleles model, pairs of individuals were considered identical-by-descent if they shared the same allele. The probability of identity-by-descent in each distance class depends on the mutation rate; the probability will be greater when there are fewer alleles. For more consistent results that are nearly independent of mutation rate, the probability of identity is often calculated as a ratio that measures genetic similarity (or differentiation) relative to a particular reference group. We calculated the kinship coefficient which measures the correlation of genetic similarity between pairs of individuals a certain distance apart relative to the genetic similarity in the whole sample.

$$F_r = \frac{p_{ij} - \bar{p}}{1 - \bar{p}} \approx \frac{E[T] - E_{ij}[T]}{E[T]}. \tag{1}$$

Here $p_{ij}$ is the probability of identity-by-descent between haploid individuals $i$ and $j$ at distance $r$ and $\bar{p}$ is the probability of identity-by-descent between random haploid individuals in the current sample (*Hardy & Vekemans, 1999*). The kinship coefficient is related to differences in the expected coalescent times, $T$, between a specific pair of individuals and a random pair in the population (*Barton et al., 2013*). Kinship coefficients were calculated for each transect and then averaged across transects for each distance class. Since this statistic is highly dependent on the sampling scheme, we sampled the same transect in all simulations.

We also calculated the $a_r$ parameter of *Rousset (2000)*:

$$a_r = \frac{p_0 - p_{ij}}{1 - p_0} \tag{2}$$

which measures genetic differentiation over distance relative to the probability of identity-by-descent within a location. The $a_r$ parameter is independent of sampling scheme, but it does depend on the level of local identity-by-descent, $p_0$, in the population such that $a_r$ approaches infinity as $p_0$ approaches one (*Vekemans & Hardy, 2004*). Typically, $p_0$ is estimated from the amount of autozygosity in the population; however, we estimated $p_0$ as the probability that an individual shared an allele with one of the offspring that it competed with for the cell, which is suitable for haploid organisms and better fits its definition (*Vekemans & Hardy, 2004*).
For each simulation, we calculated the average number of unique alleles in a 50-individual transect ($\bar{k}$) and the average squared distance between parents and offspring ($2s^2$). Using $\bar{k}$, we estimated the population-level diversity, $\hat{\theta}_k$ (*Ewens, 2004* eq. 9.32) and estimated effective haploid population size as $\hat{N}_e = \hat{\theta}_k/2\mu$ and effective density as $\hat{D}_e = \hat{N}_e/A$, where $A = 10,000$.

Finally, we estimated neighborhood size using two different methods. First we used our estimated demographic parameters to calculate neighborhood size as the product $\hat{N}_b = 4\pi s^2 \hat{D}_e$. We calculated an estimate from samples from each population and calculated an average over all populations. We then estimated neighborhood size using the regressions of both $F_r$ and $a_r$ on the log of distance. The slope of the $a_r$ regression is an estimate of $1/2\pi\sigma^2 D_e$ and the slope of $F_r$ regression is an estimate of $-(1-F_0)/2\pi\sigma^2 D_e$ (*Barton et al., 2013*; *Hardy & Vekemans, 1999*; *Rousset, 2000*). We performed the regression for distance classes between 5 and 35. We estimated the slope from each population sample then pooled the data from all the samples to get a combined slope estimate.

## RESULTS

### Behavior of dispersal distributions

The more leptokurtic distributions (exponential, gamma-1 and Lomax) with a high probability peak near zero have a much higher probability of not dispersing from the original cell, especially when $\sigma$ is low. The Pareto distribution, which has a fat tail but has been shifted so it does not have a peak at zero, has a very low probability of not dispersing. Under the gamma distribution as the $\alpha$ parameter increases, the probability of remaining at the origin decreases; when $\alpha = 8$ the probability is nearly zero for all values of $\sigma$. Figure S2 shows the empirical cumulative distributions generated from 10,000 simulated dispersal events from each distribution. The probability of not dispersing from the original cell is indicated by the height of the left-most horizontal line for each distribution.

The average squared parent–offspring dispersal distance, $s^2$, observed for each distribution was very similar with a relative error of less than 5% from the expected $\sigma^2$ value (Table 1); however, the distribution of these values over sampled generations varied (Fig. S3A). Expectedly, the thin tailed or no-tail (triangular) dispersal distributions have the smallest variance because their properties are easier to represent with a small number of samples. The Lomax distribution has the highest variance with the median falling slightly below the expected value.

Figure S3B shows the distribution of the average cubed parent–offspring dispersal distances, $s^3$, for each transect. The theoretical third moment of the Lomax and Pareto distributions is infinite, while it is not possible to simulate this on a finite landscape, we do observe values of $s^3$ that are several orders of magnitude larger than distributions with finite third moments. The distribution of $s^2$ and $s^3$ for the Lomax and Pareto distributions both have a large positive skew.

### Allelic diversity

The distribution of the number of unique alleles is similar for most of the dispersal kernels with the median falling near the expected value under the infinite alleles model

**Table 1 Estimated neighborhood sizes are similar across all dispersal distributions.** Estimates of allele diversity, $\hat{\theta}_k$, effective population density, $\hat{D}_e$, dispersal, $s^2$, and neighborhood size. Neighborhood size is estimated two different ways. $\hat{N}_{b(\theta)}$ is $4\pi s^2 \hat{D}_e$ where $\hat{D}_e$ is estimated from $\hat{\theta}_k$. $\hat{N}_{b(a_r)}$ is twice the inverse of the slope of $a_r$ and the log of distance. The expected neighborhood size ($4\pi\sigma^2 \cdot 1$) is 12.56, 28.28, 50.26, and 201.06 for $\sigma = 1$, 1.5, 2, and 4, respectively.

| | $\hat{\theta}_k$ | $\hat{D}_e$ | $s^2$ | $\hat{N}_{b(\theta_k)}$ | $\hat{N}_{b(a_r)}$ | $\hat{\theta}_k$ | $\hat{D}_e$ | $s^2$ | $\hat{N}_{b(\theta_k)}$ | $\hat{N}_{b(a_r)}$ |
|---|---|---|---|---|---|---|---|---|---|---|
| | | | **1** | | | | | **1.5** | | |
| Ray | 1.82 | 0.91 | 0.99 | 11.31 | 13.07 | 1.83 | 0.91 | 2.33 | 26.79 | 31.16 |
| Exp | 2.09 | 1.04 | 1.04 | 13.70 | 14.32 | 2.04 | 1.02 | 2.26 | 29.04 | 29.00 |
| Nor | 1.94 | 0.97 | 0.98 | 11.94 | 13.49 | 1.91 | 0.95 | 2.31 | 27.69 | 30.61 |
| Tri | 1.82 | 0.91 | 1.00 | 11.37 | 13.58 | 1.83 | 0.92 | 2.36 | 27.18 | 31.02 |
| Gam 1 | 2.07 | 1.04 | 1.05 | 13.63 | 14.41 | 2.01 | 1.00 | 2.32 | 29.22 | 30.07 |
| Gam 2 | 1.89 | 0.94 | 0.98 | 11.62 | 12.80 | 1.85 | 0.92 | 2.32 | 26.98 | 30.13 |
| Gam 4 | 1.83 | 0.92 | 1.00 | 11.49 | 12.88 | 1.87 | 0.94 | 2.32 | 27.31 | 28.27 |
| Gam 8 | 1.80 | 0.90 | 1.01 | 11.45 | 13.31 | 1.79 | 0.90 | 2.32 | 26.16 | 29.91 |
| Lom 2.4 | 2.97 | 1.49 | 1.06 | 19.70 | 13.41 | 2.62 | 1.31 | 2.16 | 35.53 | 26.65 |
| Lom 2.6 | 2.88 | 1.44 | 0.97 | 17.61 | 13.23 | 2.47 | 1.24 | 2.34 | 36.25 | 26.11 |
| Lom 2.8 | 2.73 | 1.36 | 1.04 | 17.78 | 12.82 | 2.41 | 1.21 | 2.22 | 33.66 | 25.30 |
| Lom 3 | 2.72 | 1.36 | 1.00 | 17.07 | 14.28 | 2.36 | 1.18 | 2.34 | 34.71 | 28.50 |
| Par 2.4 | 1.98 | 0.99 | 0.98 | 12.18 | 11.71 | 1.93 | 0.97 | 2.19 | 26.56 | 27.12 |
| Par 2.6 | 1.95 | 0.98 | 1.04 | 12.74 | 13.82 | 1.81 | 0.91 | 2.28 | 25.98 | 27.95 |
| Par 2.8 | 1.90 | 0.95 | 0.97 | 11.57 | 12.25 | 1.85 | 0.93 | 2.25 | 26.16 | 30.85 |
| Par 3 | 1.89 | 0.95 | 0.99 | 11.80 | 13.56 | 1.89 | 0.94 | 2.24 | 26.54 | 29.79 |
| | | | **2** | | | | | **4** | | |
| Ray | 1.97 | 0.99 | 4.07 | 50.39 | 58.81 | 2.02 | 1.01 | 16.11 | 204.93 | 236.23 |
| Exp | 2.02 | 1.01 | 4.08 | 51.88 | 49.60 | 2.09 | 1.05 | 16.16 | 212.48 | 154.94 |
| Nor | 1.95 | 0.97 | 4.08 | 49.87 | 55.00 | 2.04 | 1.02 | 16.04 | 205.76 | 189.69 |
| Tri | 1.94 | 0.97 | 4.11 | 50.13 | 54.57 | 2.09 | 1.04 | 16.09 | 210.87 | 245.02 |
| Gam 1 | 2.03 | 1.01 | 4.06 | 51.74 | 52.25 | 2.16 | 1.08 | 16.15 | 218.67 | 257.28 |
| Gam 2 | 1.89 | 0.95 | 4.12 | 48.88 | 54.39 | 2.02 | 1.01 | 16.08 | 204.41 | 214.04 |
| Gam 4 | 1.94 | 0.97 | 4.08 | 49.80 | 55.60 | 1.98 | 0.99 | 15.94 | 197.97 | 191.02 |
| Gam 8 | 1.89 | 0.94 | 4.06 | 48.21 | 52.47 | 2.02 | 1.01 | 16.11 | 203.96 | 231.04 |
| Lom 2.4 | 2.48 | 1.24 | 3.98 | 62.01 | 47.94 | 2.19 | 1.09 | 16.06 | 220.82 | 180.03 |
| Lom 2.6 | 2.36 | 1.18 | 3.94 | 58.49 | 48.10 | 2.15 | 1.07 | 15.45 | 208.62 | 219.14 |
| Lom 2.8 | 2.27 | 1.13 | 4.16 | 59.23 | 51.08 | 2.14 | 1.07 | 15.81 | 212.44 | 241.05 |
| Lom 3 | 2.24 | 1.12 | 3.97 | 56.05 | 47.23 | 2.07 | 1.04 | 16.55 | 215.21 | 211.19 |
| Par 2.4 | 1.93 | 0.97 | 4.13 | 50.12 | 48.20 | 2.03 | 1.02 | 16.04 | 204.74 | 192.65 |
| Par 2.6 | 1.95 | 0.97 | 4.11 | 50.29 | 51.74 | 2.03 | 1.02 | 15.91 | 203.23 | 189.19 |
| Par 2.8 | 1.98 | 0.99 | 4.02 | 49.95 | 47.73 | 1.95 | 0.97 | 15.53 | 189.90 | 219.90 |
| Par 3 | 1.98 | 0.99 | 4.10 | 50.92 | 49.58 | 2.01 | 1.00 | 16.30 | 205.48 | 169.53 |

(Fig. S4). The expected number of alleles under the infinite alleles model is equal to $\sum_{i=0}^{n-1} \theta/(\theta + i) = 7.03$ where $n = 50$ is the number of individuals in the sampled transect. The Lomax distributions have a higher median number of alleles at lower values of $\sigma$ but this gets closer to the expected value when $\sigma > 2$. The average diversity is also slightly elevated for the exponential and gamma-1 simulations.

Differences in effective population size between simulations can be measured by comparing the number of unique alleles observed in the transects. Different dispersal kernels produce similar levels of diversity, except for the Lomax distributions which have a higher $\theta_k$ and consequently a larger effective population size (Table 1).

**Spatial autocorrelation and isolation-by-distance**

To describe the patterns of isolation-by-distance, we first measured the average probability of identity-by-descent for each sampled population as a function of distance. All dispersal kernels produced very similar patterns of isolation-by-distance especially for larger distance classes (Fig. 1). The probability of identity-by-descent is higher at small distance when $\sigma$ is small but the relationship flattens out when $\sigma = 4$. Differences between the different dispersal distributions become apparent when the distance between individuals is small. The more leptokurtic dispersal distributions have a steeper incline as distance decreases and they have a higher probability of autozygosity at distance class zero. The plots for the triangular distribution nearly perfectly overlap the plots for the Rayleigh distribution in all cases.

Because the probability of identity-by-descent is sensitive to differences in the number of alleles present in the sample, we also calculated the pairwise-kinship coefficient over the log of distance (Fig. 2). The kinship coefficient shows how much more or less similar pairs of individuals in a given distance class are compared to the sample as a whole. The kinship coefficient is nearly independent of differences in allele number and there is much better overlap of the plots for the different dispersal distributions. When the kinship coefficient is plotted against the log of distance there is a negative linear relationship over a certain range of distances (*Hardy & Vekemans, 1999*). The slope of this linear range is also similar across distributions for each value of $\sigma$.

Finally, we plot *Rousset (2000)*'s $a_r$ parameter against the log of distance. There is a positive linear relationship between $a_r$ and the log of distance (Fig. 3). The slope of $a_r$ is fairly similar among the dispersal distributions for a given value of $\sigma$. However, there is less overlap in the plots for the different dispersal distributions because the overall amount of differentiation varies.

**Estimated neighborhood size**

The $\hat{N}_{b(\theta_k)}$ estimates are shown in Table 1 and Fig. 4A. Table 1 shows the average estimate over all population samples. The colored dots in Fig. 4A show this same average relative to the expected values and the bars represent the middle 50% of the individual sample estimates. As mentioned previously, the populations with Lomax dispersal tend to have a greater number of unique alleles and this translates to higher $\hat{\theta}_k$, higher effective population size, and ultimately higher effective density. The estimates for $s^2$ were highly variable but

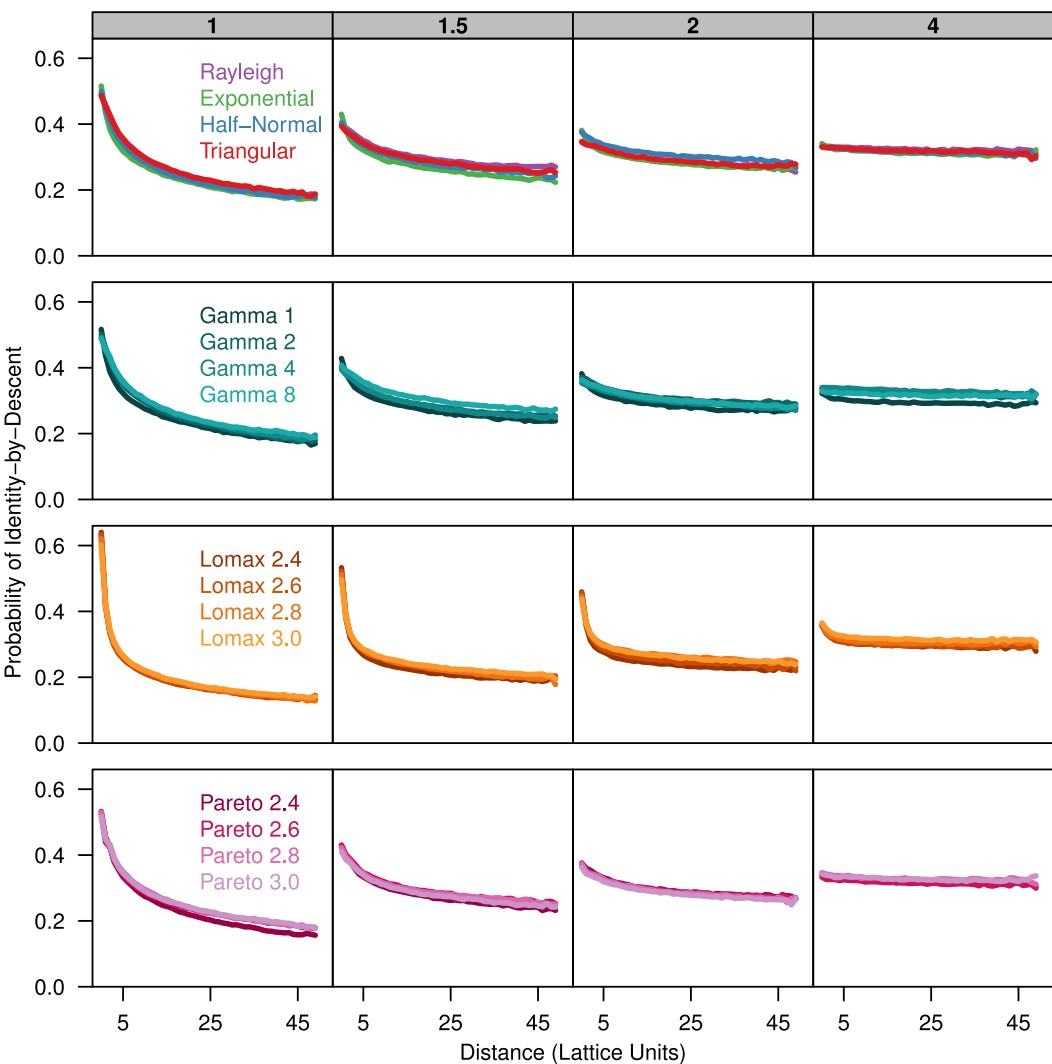

**Figure 1 Identity-by-descent is similar between different dispersal models.** Each plot shows the average probability of identity-by-descent for pairs of individuals in each distance class. Each panel represents simulations run with different $\sigma$ parameters (gray box) for different groups of dispersal distributions.

skewed towards lower values. As a result, the estimates of $\hat{N}_{b(\theta_k)}$ for the Lomax distribution appear to be higher on average but the estimates are skewed. Otherwise, the estimates for the other dispersal distributions are similar and close to the expected values.

Table 1 shows the $\hat{N}_{b(a_r)}$ estimates calculated as the twice the inverse of the regression of $a_r$ and the log of distance for the pooled sample data. Estimates using the slope of the $F_r$ statistics were identical so they are not shown. The colored dots in Fig. 4B show the slope estimate of the combined data relative to the expected slope and the bars represent the middle 50% of the slopes from individual populations. All of the dispersal distributions have similar slopes. When $\sigma = 4$, the actual spread of the slope values is smaller than the the spread of the slopes for the other values of $\sigma$ (not shown), but in Fig. 4 the values are relative so the middle 50% is wider.

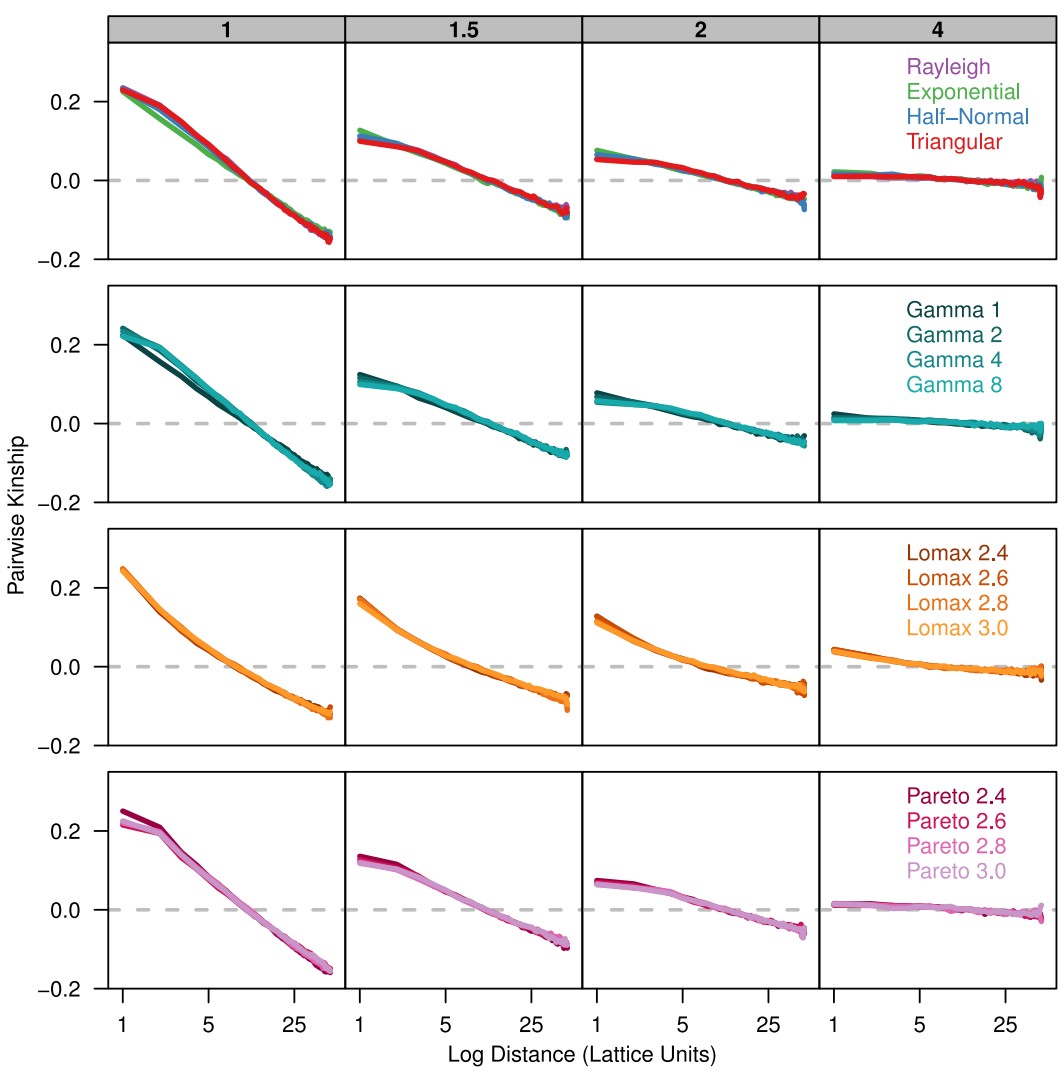

**Figure 2  Kinship coefficients are similar between different dispersal models.** Each plot shows the average kinship coefficient for pairs of individuals over the log of the distance between them. Each panel represents simulations run with different $\sigma$ parameters (gray box) for different groups of dispersal distributions. The gray dashed line is at zero so values above the line are more similar than the sample as a whole while values below the line are less similar than the population as a whole.

## DISCUSSION

Approximating continuous dispersal on a discrete lattice will introduce obvious biases when the dispersal distance is small compared to the scale of the lattice nodes (*Chipperfield et al., 2011*). This bias can be seen in Fig. S2 by the jagged nature of the empirical cumulative distribution (ECDF) (especially when $\sigma$ is small) compared to the CDF of the continuous distribution. In the simulation, the distance between nodes is one lattice unit so dispersal has to exceed at least a distance of 0.5 lattice units to leave the original cell. For Lomax simulations with small $\sigma$, the high probability density near zero falls rapidly before a distance of 0.5 lattice units has been reached. This means that the majority of dispersal

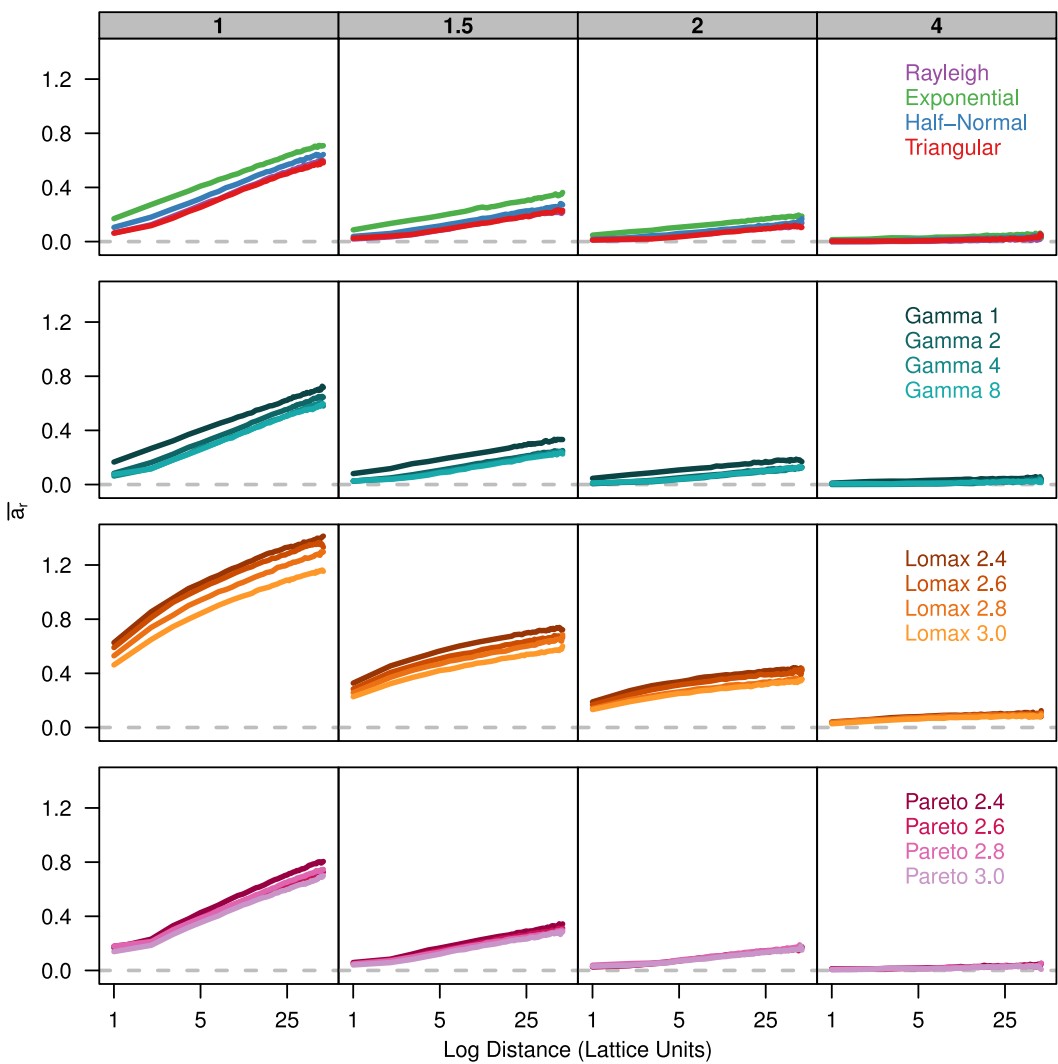

**Figure 3** **Slopes of genetic differentiation are similar between different dispersal models.** Each plot shows the average differentiation, $a_r$, for pairs of individuals over the log of the distance between them. Each panel represents simulations run with different $\sigma$ parameters (gray box) for different groups of dispersal distributions.

events do not leave the parent cell. The Pareto and Lomax distributions share a similar shape and a wide tail, but unlike the Lomax distribution, the mode of the Pareto is greater than zero and almost all dispersal events leave the original cell. We refer back to the differences between the Lomax and the Pareto when we discuss whether we can differentiate results that are specific to dispersal with a high peak at zero or are more general to wide-tailed dispersal.

Allelic diversity is near the expected value predicted by the infinite alleles model for most distributions. The Lomax distributions tend to have a higher number of alleles up until $\sigma = 4$. This appears to be in agreement with *Maruyama (1972)* which showed that the effective population size is larger than the census size when $\sigma < 1$ which is the case in many of the Lomax simulations (Fig. S3). Because the median allele number for the Pareto

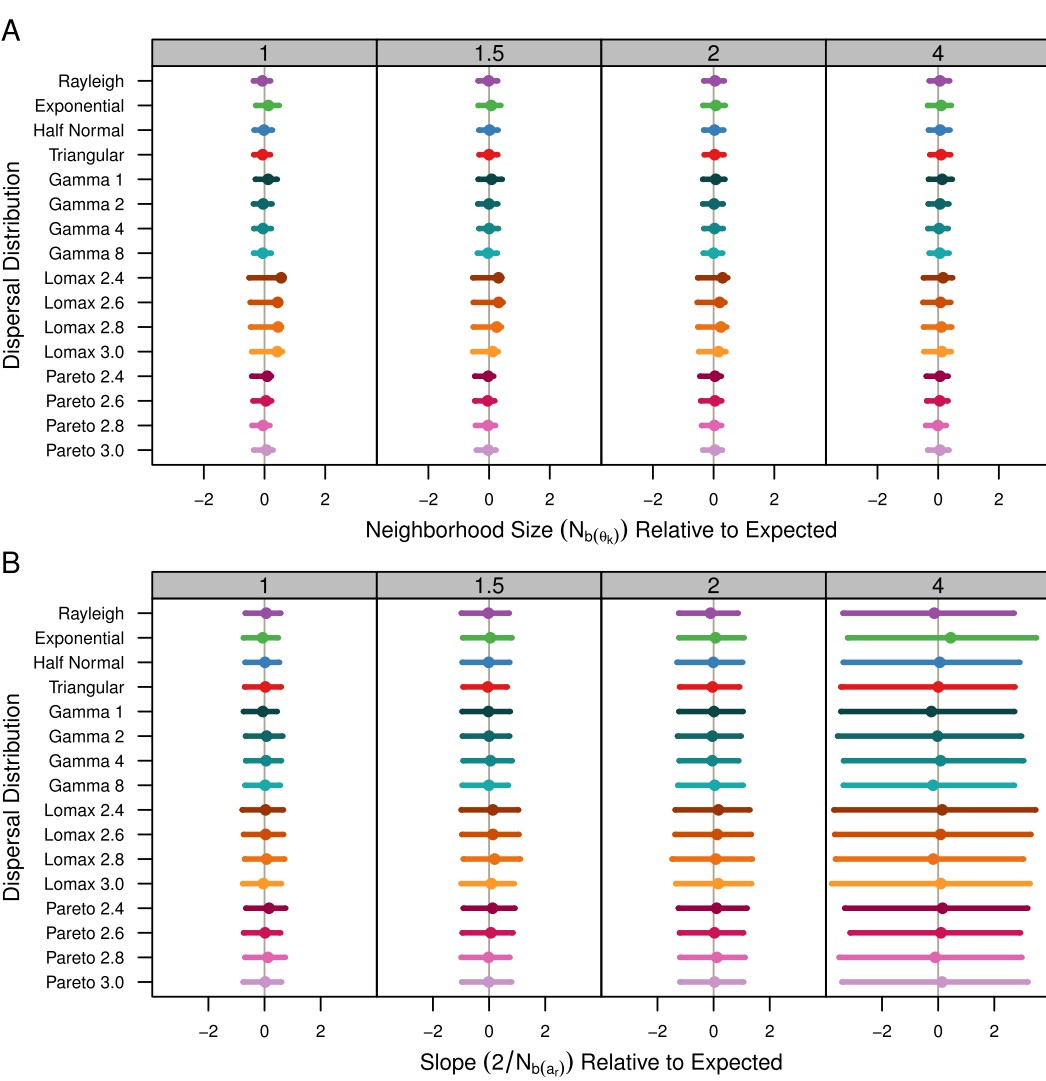

**Figure 4** **Estimated neighborhood sizes are similar across all dispersal distributions.** Neighborhood size is estimated two different ways. (A) $N_{b(\theta_k)}$ is $4\pi s^2 \hat{D}_e$ where $\hat{D}_e$ is estimated from $\hat{\theta}_k$. The dot is the average from all populations samples and the bars represent the middle 50% of estimates from individual samples. (B) The slope estimates, $\frac{2}{N_{b(ar)}}$, of $a_r$ and the log of distance. The dots represent the slope estimate from the combined data from all samples and the bars represent the middle 50% of slopes from individual samples.

simulations falls near the expected value, it seems likely that the higher allelic diversity in the Lomax simulations is due to the high probability of not dispersing. This is supported by the fact that the average diversity is slightly higher for the exponential and gamma-1 as well. When dispersal is unlikely to occur outside of the original cell, the number of migrants is low and the pool of offspring before competition will consist mostly of offspring from the same parent. It is unlikely that migrants will become established at their new location after competition and thus more alleles will be maintained.

Much of the theory of isolation-by-distance in continuous populations is based on infinite or periodic lattice models. Here we simulated dispersal in a continuous population

occupying a finite lattice with absorbing boundaries to better understand the effect of the dispersal kernel on isolation-by-distance models on a more natural landscape. As expected under isolation-by-distance, the probability of identity-by-descent between neutral alleles in pairs of individuals decreases as the distance between them increases. When neighborhood size is small, the relationship is very pronounced with a high initial probability that quickly declines. As neighborhood size increases ($\sigma = 4$), this relationship nearly disappears. This is similar to two-dimensional stepping stone models that show strong differentiation between populations when $Nm \ll 1$ and little differentiation when $Nm > 4$ (*Kimura & Maruyama, 1971*).

Simulations with our different dispersal kernels show a strikingly similar pattern of isolation-by-distance. However, theory predicts that when distance is small, deviation in the shape of the dispersal kernel relative to the Rayleigh distribution will become important (*Rousset, 1997*; *Rousset, 2000*). This is evident in our results when we compare the probabilities of identity-by-descent at small distances between the different dispersal kernels. When the dispersal kernel is leptokurtic, the probability is higher between individuals occupying the same location and it is slightly lower for short distances compared to the Rayleigh results. The pattern of identity-by-descent in other distributions, including the triangular are nearly identical to the Rayleigh. The situation is similar for the pairwise kinship except there is even greater similarity between the different dispersal kernels.

*Rousset (2008)* makes it clear that the increase of genetic differentiation with distance is robust to the shape of the dispersal kernel but the overall magnitude of differentiation will depend on the shape of the kernel. Looking at the relationship between $a_r$ and the log of distance for our simulations, we can see that the slope for each distribution is similar for larger distance values but the plots are shifted up or down depending on kurtosis. Compared to the other wide tailed distributions, the Pareto distribution is not shifted upward due to the lack of dispersal at the origin. The $a_r$ statistic is a ratio that compares the amount genetic differentiation between individuals at certain distance to the differentiation within a single individual. When the probability of identity-by-descent within an individual is high, the differentiation between neighbors will appear much higher due to a steep initial drop in identity. As a result, the $a_r$ statistic will be greater for leptokurtic distributions even if the actual probability of identity is similar to other distributions.

As expected, the neighborhood-size estimates are similar to the expected value for all simulations. Neighborhood size was slightly higher for the Lomax simulations when using allele diversity to estimate effective density. Otherwise, the slopes of the regression methods were similar and thus predicted similar neighborhood sizes. This reconfirms that neighborhood size is a robust descriptor of the decrease of genetic identity with distance. It also seems clear that fat-tailed dispersal kernels do not have much of an effect in isolated continuous populations.

The triangular distribution has not been considered a reasonable distribution to use for modeling biological dispersal. However, as discussed previously, it arises from the simple assumption that dispersal is locally panmictic, making it potentially useful. When we compared the triangular distribution against more popular dispersal models, there were no qualitative differences between the resulting patterns of isolation-by-distance.
**Table 2  Triangular dispersal algorithm is the most efficient.** Execution time and relative time for $10^9$ dispersal events from different dispersal functions ordered from most to least efficient.

| Dispersal function | CPU seconds | Relative time |
| --- | --- | --- |
| Triangular | 21.853 | 1 |
| Rayleigh | 27.713 | 1.268 |
| Exponential | 106.434 | 4.870 |
| Half Normal | 106.771 | 4.886 |
| Gamma | 119.357 | 5.462 |
| Pareto | 127.218 | 5.822 |
| Lomax | 127.376 | 5.829 |

The triangular dispersal model can serve as a null model for the probability that two lineages will meet and coalesce in a previous generation. Identity-by-descent may be defined as the total probability of coalescence between the current generation, $t_0$, and a generation at some time $t$ in the past (*Rousset, 2002*). When a population is not panmictic due to limited dispersal, the time to coalescence depends on the probability that the two lineages will move close enough together so that there is some probability that they shared a parent in the previous generation. When the dispersal kernel has an infinite tail, there is always some small probability that two individuals coalesce even if they are very far apart. Because the triangular distribution is finite with a maximum distance of $2\sigma$, the probability that two individuals coalesce in the previous generation is $1/(4\pi\sigma^2 D)$ if they are separated by a distance less than $2\sigma$ and zero otherwise.

The triangular distribution allows us to simulate dispersal more efficiently than other dispersal kernels because it is uniform over a finite area. It allows us to easily pre-compute probabilities of dispersal to neighboring cells and use an efficient discrete sampling algorithm to sample dispersal positions. A similar approach is possible for other dispersal distributions. For distributions with infinite tails this would require defining a truncated distribution which captures the bulk of the dispersal probabilities. Then, for two dimensions, double integrals would need to be calculated to determine the probabilities of dispersal to locations on the lattice. These pre-computations are laborious because in addition to the double integrals, many cells will have non-zero probabilities. For the triangular distribution, only cells in a radius of $2\sigma$ will have non-zero probability and since the distribution is uniform, the probabilities are easy to calculate.

Our results suggest that the relationship between probability of identity-by-descent and distance is similar for a wide range of dispersal kernels in a continuous population, and both theoretical and computational concerns suggest that triangular distributions should be included in the molecular ecologist's toolkit. However, these results should not be taken to mean that it is always safe to ignore the shape of the dispersal kernel. As we demonstrate here, the high number of extremely limited dispersal events under the Lomax distribution increases the probability of identity-by-descent within a cell. In a hermaphroditic plant this could translate into a higher rate of self-fertilization. The shape of the tail can impact the number of long distance dispersal events which may affect

the rate of population expansion, colonization, responses to climate change, population fragmentation and the movement of genes between locally adapted populations. Each of these processes will be affected by the dispersal distribution chosen for the simulation. However, when simulating a population structured by isolation-by-distance, the shape of the dispersal kernel does not appear to have a strong effect in a finite, isolated population. Because speed is an important factor in deploying isolation-by-distance simulations in analytical contexts, e.g., approximate Bayesian computation, we recommend using the triangular distribution when long distance dispersal and other features of the dispersal kernel can safely be ignored.

## ACKNOWLEDGEMENTS

The authors would like to thank R Schwartz, D Winter and S Wu for helpful comments, and K Dai for programming tips. The authors would also like to thank Jeffrey Ross-Ibarra, Michael Blum, and one anonymous reviewer for their suggestions which improved this manuscript.

## APPENDIX A. TRIANGULAR DISTRIBUTED DISTANCES CAN PRODUCE A UNIFORM DISTRIBUTION ON A DISK

### Proof

The probability density of a uniform distribution over a finite two-dimensional shape is defined as:

$$f(x,y) = \begin{cases} \dfrac{1}{\text{area of } S} & \text{if } (x,y) \in S \\ 0 & \text{otherwise} \end{cases}$$

where $(x,y)$ are coordinates on the Cartesian plane and $S$ is the set of all points within the shape. A uniform distribution on the region bounded by a circle is defined by $\frac{1}{\pi R^2}$ where $R$ is the radius of the circle. We are interested in a circle with radius $R = 2\sigma$ and area $A = 4\pi\sigma^2$ so the non-zero part of the joint probability distribution is given by:

$$f(x,y;\sigma) = \frac{1}{4\pi\sigma^2} \quad \text{when } x^2 + y^2 \leq 4\sigma^2.$$

Using the change of variables theorem for polar coordinates, $(r,\theta)$, we have:

$$\iint_D f(x,y) \, dx \, dy = \iint_{D^*} f(r\cos\theta, r\sin\theta) r \, dr \, d\theta$$
$$= \iint_{D^*} \frac{r}{4\pi\sigma^2} \, dr \, d\theta$$
$$= \iint_{D^*} \frac{1}{2\pi} \frac{r}{2\sigma^2} \, dr \, d\theta.$$

We then integrate out the angle $\theta$ to isolate the distribution of distance, $f(r;\sigma)$:

$$f(r;\sigma) = \int_0^{2\pi} \frac{r}{4\pi\sigma^2} \, d\theta = \frac{r}{4\pi\sigma^2} \theta \Big|_0^{2\pi} = \frac{r}{2\sigma^2} \quad \text{for } 0 \leq r \leq 2\sigma.$$

The distribution of distances is equivalent to a special case of the triangular distribution. The probability density function for the triangular distribution is

$$f(r;a,b,c) = \begin{cases} 0 & \text{for } r < a \\ \dfrac{2(r-a)}{(b-a)(c-a)} & \text{for } a \leq r \leq c \\ \dfrac{2(b-r)}{(b-a)(b-c)} & \text{for } c \leq r \leq b \\ 0 & \text{for } r > b \end{cases}$$

where $a$ is the lower limit, $b$ is the upper limit, and $c$ is the mode. In the special case, we set $a = 0$ and $b = c = 2\sigma$. The probability density function for the special case of the triangular distribution simplifies to

$$f(r;\sigma) = \begin{cases} \dfrac{r}{2\sigma^2} & \text{for } 0 \leq r \leq 2\sigma \\ 0 & \text{otherwise} \end{cases}$$

## APPENDIX B. XORSHIFT RANDOM NUMBER GENERATOR

Xorshift is a type of pseudo-random number generator that relies on exclusive-or and bitshift operators (*Marsaglia, 2003*). Xorshift is one of the most efficient, high-quality random-number generators known. Our implementation is a 64-bit xorshift with shift parameters (5, 15, 27) added to a Weyl series to decrease bit correlations (*Brent, 2007*). It passes the BigCrush tests in the TestU01 suite (*L'Ecuyer & Simard, 2007*).

## APPENDIX C. GENERATING FROM A CONTINUOUS TRIANGULAR DISTRIBUTION

Inverse sampling can be used to generate values from a triangular distribution. Note that we are only working with monotonically increasing triangular distributions and not more general formulations. If $u$ is uniformly distributed in $(0,1)$, the value $d = 2s\sqrt{u}$ has a triangular distribution with parameter $s$. However, a modified rejection sampling algorithm is faster. If $u_1$ and $u_2$ are independent and uniformly distributed in $(0,1)$, then $d = 2s\max(u_1, u_2)$ also has a triangular distribution. Because we can generate 32-bit values for both $u_1$ and $u_2$ from a single 64-bit random number, this second algorithm is more efficient than the first. While it is possible to construct a ziggurat algorithm (*Marsaglia & Tsang, 2000b*) for a triangular distribution, our second algorithm is more efficient because it involves fewer steps and never rejects.

We compared the speed of these algorithms and a naive rejection sampler using the medium Crush tests (*L'Ecuyer & Simard, 2007*). This allowed us to compare the speeds of these algorithms in a data-intensive application as well as verify that the algorithms produced independent and identically distributed values from the correct distribution. The 'maximum' algorithm took 1,656 s to complete, while the 'sqrt' took 1,700 s and the

rejection sampler took 1,911 s. The maximum algorithm produced faster execution, but only sped up the tests by 3% over sqrt.

## APPENDIX D. GENERATING DISCRETE TWO-DIMENSIONAL DISPERSAL FROM A TRIANGULAR DISTRIBUTION

We can use the maximum algorithm above to generate the values in polar coordinates and convert them to Cartesian coordinates; however, this requires calculating sine and cosine functions, which we would rather not do. When modeling dispersal on a lattice, the bounded nature of the triangular distribution allows dispersal to be modeled discretely. To discretize this distribution on a rectangular lattice we must determine the probabilities for each cell which are proportional to the area of the cell that is covered by a disk of radius $r = 2\sigma$ (centered on a focal cell). The algorithm described here produces probability tables by calculating the appropriate area for each cell and dividing by the total area. We assume that cells are squares with unit area.

Since the disk is symmetrical, this algorithm may be simplified by calculating areas for quadrant I of the disk and mirroring those values to the other quadrants. We further simplify by calculating approximately half of the areas for quadrant I and mirroring those as well. Note that this results in cells along the $x$ and $y$ axes having an area of 1/2. Starting at the center of the focal cell ($y_0 = 0$), we record the top/bottom boundary of each cell along the $y$-axis up to the radius: $y_1 = 0.5$, $y_2 = 1.5, \ldots$, $y_n = n - 0.5$ where $n = \sup_{n \in \mathbb{Z}} y_n \leq r$.

Next we calculate the area of the first column of cells which has a left boundary at $x_0 = 0$ and a right boundary at $x_1 = \min(0.5, r)$:

$$A = \int_{x_0}^{x_1} \sqrt{r^2 - x^2}\, dx.$$

Starting with the bottom cell, we check if the area of a cell is less than the area of the column. If so, the cell is completely contained in the disk, and the cell is assigned a weight equal to its area. Its area is then subtracted from the area of the column. We continue this procedure until the the area of last cell is less than the remaining area of the column and assign the final cell a weight equal to the remaining area in the column.

We then move to the next column by setting $x_0 = 0.5$ and $x_1 = 1.5$. However, before we calculate the area, we must check if the edge of the disk passes through the bottom of the top cell. This occurs if $x_1^2 + y^2 > r^2$, where $y$ is the value of the bottom boundary of the cell. When this occurs, we split the column into two smaller columns and each column is processed just like before. We continue calculating the area of subsequent columns until we reach the column that contains the point $\{x, y\} = \left\{ r/\sqrt{2}, r/\sqrt{2} \right\}$, which marks the point where the edge of the disk intersects the diagonal. After this column is processed, the weights for these cells can be copied symmetrically. The weight of each cell is divided by the total area of the disk and becomes a probability. These probabilities are then copied symmetrically to the other three quadrants. The completed table of probabilities can then be passed into the alias algorithm for discrete sampling (*Vose, 1991*).

Our implementation of a discretized triangular kernel can be found in src/disk.h and src/disk.cpp in the source code. Code for generating an alias table can be found in src/aliastable.h.

# APPENDIX E. RELATIVE EXECUTION TIME OF DISPERSAL FUNCTIONS

To compare the run time for the different dispersal functions we simulated one dispersal event from each cell on a $100 \times 100$ landscape 100,000 times for a total of $10^9$ dispersal events. For each simulation $\sigma = 1$, and $\alpha = 3$ for the two parameter distributions. The CPU time was averaged over 5 different runs (Table 2). Our implementation of the triangular distribution was the most efficient followed by the Rayleigh which took about 26.8% longer on average. The half-normal and the exponential functions had similar execution times but took nearly 5 times longer than the triangular function. The gamma, Pareto, and Lomax were the least efficient functions running over 5 times longer than the triangular function.

### Funding

The authors received funding from the School of Life Sciences, Arizona State University for this work. The funder had no role in study design, data collection and analysis, decision to publish, or preparation of the manuscript.

### Grant Disclosures

The following grant information was disclosed by the authors:
School of Life Sciences, Arizona State University.

### Competing Interests

Reed A. Cartwright is an Academic Editor for PeerJ.

### Author Contributions

- Tara N. Furstenau conceived and designed the experiments, performed the experiments, analyzed the data, contributed reagents/materials/analysis tools, wrote the paper, prepared figures and/or tables, reviewed drafts of the paper.
- Reed A. Cartwright conceived and designed the experiments, contributed reagents/materials/analysis tools, wrote the paper, reviewed drafts of the paper.

### Data Availability

Source code for simulations can be found at https://github.com/tfursten/IBD/tree/vpub. Simulation parameters and results can be accessed at 10.6084/m9.figshare.1611097.

### Supplemental Information

Supplemental information for this article can be found online at http://dx.doi.org/10.7717/peerj.1848#supplemental-information.

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
