# Peer review of "The effect of the dispersal kernel on isolation-by-distance in a continuous population"

_PeerJ, doi:10.7717/peerj.1848_

## Round 0.1 · original submission · Major Revisions

Both reviewers think your article is eventually suitable for publication in PeerJ. Reviewer 1, however, notes some concerns about clarifying the method and their point #4 is worth considering/justifying. Reviewer #2 believes, as do I, that more could probably be done to place this work in context, emphasizing what the new contributions are.

Reviewer 1 ·

Basic reporting

Overall the article was well-written and sufficiently introduces the topic and its importance.

Experimental design

It is not clear how many simulation replicates were done for each dispersal kernel. This should be clarified by the authors. Along with my comments below, I think the simulation set-up requires further testing as described in my major comment 4.

Validity of the findings

The findings presented seem valid, but I would like to see them expanded, as described in my major comments, before the claim that these distributions create equal patterns is fully supported.

Additional comments

Summary:
This study compares the effects of different types of dispersal probability distributions on isolation by distance across a simulated landscape of 10,000 individuals on a lattice grid of 100x100. They test seven different types of dispersal kernels and conclude that all generate the same signals of isolation by distance, therefore indicating that using the most computationally efficient algorithm and kernel is best to use rather than a more biologically realistic kernel. I found this study to be useful and interesting but have several questions detailed below regarding description of the triangular distribution and how biologically relevant these results may be for studies wanting to use simulations examining other processes on top of isolation by distance that may indeed behave differently under these disparate kernels.

Major Comments:
1. The abstract and lines 123-124 say that the triangular distribution has uniform dispersal probability within the neighborhood area, but this does not seem to be what is described in table 1 where the probability of dispersing increases up to the cut-off point of 2 sigma. Can this be clarified? Furthermore why did the authors choose this special case of the triangular distribution where a = 0? Does changing the a, b, and c parameters in this distribution change the results? Multiple alpha values were tested for the other distributions, so it seems fair and worthwhile to evaluate this distribution with different shape parameters.
2. Continuing on the topic of the triangular distribution, it is the only one with a stated cut-off value for maximum dispersal distance. Is there truly no cut-off for the other distributions? If these are all being discretized onto a lattice, then not cutting off the other distributions will give non-zero probabilities of dispersing across almost all, if not all, of the 100x100 grid used. Clearly this will make the other distributions take more computational time and power. I suggest implementing a cutoff at some value of sigma that still includes the bulk of the dispersal probability, but ignores non-zero probabilities past something like 6 or 8 sigma that are effectively zero probability of dispersing on average. Perhaps setting some limit such as the distance in sigma that includes 99% (or higher) of the dispersal probability.
3. I would like to see slightly more emphasis on the point made in lines 27 and 344-347 which state how the shape of the dispersal kernel can affect “many population processes at different scales” in addition to isolation by distance which is a very important point. The title does make it clear the implications are only for studying isolation by distance, but I am always wary of readers disregarding assumptions innate in a study like this.
4. 100x100 lattice with 10,000 individuals seems quite small since only one individual is allowed in each cell. Given that the smaller distances resulting in jagged cumulative distributions in Fig. 1. If I am correct in understanding that “1 sigma” means sigma equals the size of one cell’s length, then it might be more appropriate to test several larger sigma values over a larger landscape. Especially for the distributions where there is a larger probability of not dispersing, e.g. exponential, this seems like it may only be realistic for certain biological cases such as organisms that are very sparse on the landscape. But many organisms can be found at much higher densities and may also be most likely to not disperse far from their natal patch, which is not allowed in this system with the 1 individual per cell limit. Increasing sigma over a larger landscape should represent a scenario like this.

Minor Comments:
1. The figures all have many panels each of quite a small size that makes interpreting and digesting the results visually very difficult.
2. What were the computational times that each of the distributions took to run? How significant are the differences so that someone interested in using the triangular over another can decide how worthwhile this is versus sacrificing biological realism?
3. If alleles are neutral, it is entirely random who is chosen to remain in the cell after density regulation?
4. How many replicates were run per simulated dispersal distribution?

·

Basic reporting

No Comments

Experimental design

No Comments

Validity of the findings

No Comments

Additional comments

I have no major comments to make about the manuscript. The authors find that the rate at which genetic differences between individuals increases is a function of the variance of the dispersal kernel but does not depend on the shape of the dispersal kernel. Several authors have already shown this result and the authors of the reviewed paper should be more precise about what is new in the current paper. My understanding is that most cited papers provided theoretical justifications whereas the current paper confirms this finding using simulations. Other comments are more minor.

Abstract
All parameters introduced in the abstract should be defined (D_e,sigma).
“how to draw from a triangular” -> “How to draw random variables from a triangular”.

line 40 “does depends” -> “depends”

I do not think that providing raw values in Table 2 is useful. If the objective is to show that estimated neighborhood sizes are similar across distributions, a figure can do a better job than a table.

---

## Round 0.2 · Minor Revisions

While a table of other papers using the 100 x 100 is not needed in the paper, I'd like to see mention of the fact that it is common.

I'm also not totally sure that all 8 figures are needed in the main text. Perhaps some could be moved to the supplement?

---

## Round 0.3 · accepted · Accept

Looks good, thank you for the revision.